# The Complex Relation between Atrial Cardiomyopathy and Thrombogenesis

**DOI:** 10.3390/cells11192963

**Published:** 2022-09-22

**Authors:** Elisa D’Alessandro, Joris Winters, Frans A. van Nieuwenhoven, Ulrich Schotten, Sander Verheule

**Affiliations:** Department of Physiology, Cardiovascular Research Institute Maastricht, Maastricht University Medical Center, 6200 MD Maastricht, The Netherlands

**Keywords:** atrial cardiomyopathy, atrial fibrillation, thrombogenesis, cardiac remodeling

## Abstract

Heart disease, as well as systemic metabolic alterations, can leave a ‘fingerprint’ of structural and functional changes in the atrial myocardium, leading to the onset of atrial cardiomyopathy. As demonstrated in various animal models, some of these changes, such as fibrosis, cardiomyocyte hypertrophy and fatty infiltration, can increase vulnerability to atrial fibrillation (AF), the most relevant manifestation of atrial cardiomyopathy in clinical practice. Atrial cardiomyopathy accompanying AF is associated with thromboembolic events, such as stroke. The interaction between AF and stroke appears to be far more complicated than initially believed. AF and stroke share many risk factors whose underlying pathological processes can reinforce the development and progression of both cardiovascular conditions. In this review, we summarize the main mechanisms by which atrial cardiomyopathy, preceding AF, supports thrombogenic events within the atrial cavity and myocardial interstitial space. Moreover, we report the pleiotropic effects of activated coagulation factors on atrial remodeling, which may aggravate atrial cardiomyopathy. Finally, we address the complex association between AF and stroke, which can be explained by a multidirectional causal relation between atrial cardiomyopathy and hypercoagulability.

## 1. Introduction

Atrial cardiomyopathy accompanying AF is associated with thromboembolic events, such as stroke [1]. AF and stroke share many risk factors and, therefore, their causal relation involves multiple complex pathological processes. This review aims to discuss the most relevant mechanisms by which atrial cardiomyopathy preceding AF leads to thromboembolic events and to address the multidirectional causal relation between atrial cardiomyopathy and stroke.

First, we briefly discuss the relation between AF and atrial cardiomyopathy, and those animal studies that have contributed to elucidating the role of specific pathogenic stimuli on the development of different types of cardiomyopathies. Subsequently, we focus more extensively on the link between atrial cardiomyopathy and thrombogenic events. We review the most relevant atrial structural and molecular changes that predispose patients to thrombogenesis within the atrial cavity and myocardial interstitial space. From this point of view, we elaborate on the pleiotropic effect of coagulation factors and their role in atrial cardiomyopathy. Finally, we address the complex association between AF and stroke.

## 2. Atrial Cardiomyopathy and Atrial Fibrillation

Atrial fibrillation (AF) is the most common sustained tachyarrhythmia in clinical practice. The prevalence of AF rises steeply with age [1]. It has long been recognized that the risk for AF is increased by underlying structural heart disease, including coronary artery disease, prior myocardial infarction, heart failure and valvular disease [1]. This led to the distinction between ‘AF with preexisting structural heart disease’ and ‘lone AF’, i.e., AF occurring in the absence of structural heart disease. However, many other non-cardiac disease factors also increase the likelihood of AF, e.g., obesity, sleep apnea, and hyperthyroidism, in the absence of clinically detectable changes in the cardiac structure or function [1,2]. For example, diabetes mellitus has been associated with an increased risk of developing AF. The mechanisms by which this metabolic disorder would lead to AF are still under debate. Growing evidence suggests the involvement of diabetes-related oxidative stress and inflammatory state [3]. Moreover, glucose and insulin disturbances are also associated with pathological changes in the heart, as suggested by the increase in the left ventricular mass accompanying the worsening of glucose intolerance [4].

In this sense, the atria may act as the ‘coalminers canary’ of the circulation, due to its apparent sensitivity to hemodynamic and metabolic abnormalities, with AF as its most relevant clinical manifestation. Very few AF patients do not have any of the known risk factors. For this reason, the concept of ‘lone AF’, in the broader sense of idiopathic AF without underlying causative pathology, has been gradually abandoned [5].

In recent years, the term ‘atrial cardiomyopathy’ has been proposed to describe ‘any complex of structural, architectural, contractile or electrophysiological changes affecting the atria with the potential to produce clinically relevant manifestations’ [6]. In this view, many different pathologies, such as structural heart disease as well as systemic metabolic alterations, can lead to disease processes affecting the atrial myocardium, leaving a ‘fingerprint’ of structural and functional changes (Figure 1). Many of these alterations, e.g., fibrosis, myocyte hypertrophy, and fatty infiltration, can increase vulnerability to AF [7,8,9]. Initially, AF is characterized by atrial electrical remodeling followed by a much slower and irreversible process, which is structural remodeling. In fact, once AF develops, its rapid atrial rates and loss of organized contractility cause, among others, calcium overload, ischemia, oxidative stress, and stretch that further contribute to atrial electrical and structural changes [10,11,12,13]. In this context, AF can either exacerbate pre-existing remodeling processes or contribute to the new onset of pathological changes in the atria, becoming either the consequence or the cause of atrial cardiomyopathy.

Frustaci et al. reported that even in patients fitting the definition of ‘lone AF’ used at that time, histological evidence of occult atrial cardiomyopathy (i.e., myocarditis, cardiomyopathy, myolysis and necrosis) was apparent [14,15,16]. In patients undergoing cardiac surgery, Goette et al. reported that isolated atrial amyloidosis (i.e., an accumulation of atrial natriuretic peptide) correlated with an increased P wave duration and was predictive of AF, whereas overall fibrosis was not [16].

Atrial fibrosis is one of the most extensively studied aspects of atrial cardiomyopathy in patients [9]. Anné and coworkers reported that mitral valve disease, a well-recognized risk factor for AF, leads to increased atrial fibrosis, but that AF itself was not associated with increased atrial fibrosis [17]. In contrast, Platonov and colleagues reported increased fibrosis in patients with AF, and that fibrosis was more pronounced in patients with permanent AF than in patients with paroxysmal AF [18]. Both these studies assessed overall fibrosis through conventional histological staining, which is not particularly sensitive to endomysial fibrosis, i.e., fibrosis between myocytes within bundles [9]. We have recently shown that both overall and endomysial fibrosis was higher in patients with AF, but that AF complexity, a surrogate parameter for AF stability, was more strongly correlated with endomysial fibrosis [19]. 

Atrial biopsies for a detailed assessment of atrial cellular electrophysiology and atrial tissue structure are only available in a limited subset of patients, typically those undergoing cardiac surgery for valvular or coronary artery disease, and not in healthy control subjects. Many of these patients have multiple underlying diseases, often of unknown duration, that could lead to atrial cardiomyopathy. Therefore, the exact relation between AF risk factors and atrial cardiomyopathy is often difficult to ascertain in patients.

## 3. Diversity of Atrial Cardiomyopathy in Animal Models

Many of the disease entities that are associated with an increased risk of AF in patients have been investigated in animal models (Table 1). For practical reasons, these pathological factors are often applied in an intense form for a limited duration. Nevertheless, animal studies have provided valuable information on what pathogenic stimuli lead to which type of atrial cardiomyopathy.

The progressive, self-perpetuating nature of AF has been investigated in goat and dog models of AF maintained by rapid atrial pacing. The progression of AF in these models can be explained by the fact that AF causes changes in the atria that increase the stability of the arrhythmia. These changes include a shortening of the action potential duration and effective refractory period (electrical remodeling) and alterations in tissue structure (structural remodeling). The latter process includes altered connexin expression, myocyte hypertrophy and atrial fibrosis [31,32,33]. Whereas electrical remodeling takes place rapidly, within 1–2 days, structural remodeling progresses gradually over a time course of months [34,35]. The stability of AF (measured as the duration of AF episodes or as the amenability to pharmacological cardioversion) increases over a comparably slow time course of months [36,37]. The main histological correlates of this slow progression are myocyte hypertrophy, which occurs ubiquitously over the entire thickness of the atrial wall, and endomysial fibrosis, i.e., an increased thickness of collagen septa surrounding individual myocytes within bundles. The latter occurs predominantly in the outer millimeter of the atrial wall [20]. This type of fibrosis is associated with compromised transverse propagation and a more complex, 3-dimensional fibrillation pattern, that has been confirmed in AF patients [19,20,38,39]. AF in these animal models does not lead to myocyte death and a corresponding increase in replacement fibrosis [20,23,38,39,40,41].

In contrast, a dog model of congestive heart failure (CHF) induced by rapid ventricular pacing (RVP), did show a pronounced early phase of myocyte apoptosis and necrosis in the first 1–2 days, and a pronounced increase in overall (replacement) fibrosis after 3–5 weeks, a stage at which clinical signs of decompensating heart failure had developed [23,41]. Interestingly, the differences in structural remodeling between canine models of ‘lone AF’ and CHF are paralleled by differences in gene expression patterns, mainly with downregulation in the lone AF model and a more varied pattern of upregulation and downregulation in the CHF model [42].

Atrial dilatation is a strong predictor of AF, occurring as a result of numerous risk factors for AF, e.g., CHF and valvular disease, as well as from AF itself [43]. Although an increase in atrial size per se is proarrhythmic, atrial dilatation can also lead to changes in tissue structure that have been investigated in animal models [43]. Partial avulsion of the tricuspid valve and mitral valve lead to heterogeneous fibrosis in the right and left atrium, respectively [24,44]. However, valvular avulsion leads to an immediate increase in atrial size, and therefore a rapid increase in atrial stretch. In contrast, in a goat model of chronic AV block, which leads to a more gradual atrial dilatation over a time course of weeks, myocyte hypertrophy was reported, while overall fibrosis did not increase [25].

The effects of noncardiac risk factors for AF on the atrial structure have been less extensively investigated in animal models. In a sheep model of obesity, atrial conduction heterogeneity and increased AF stability were associated with fibrosis and fatty infiltration [27]. In a rat model, hypothyroidism increases atrial fibrosis, whereas hyperthyroidism leads to myocyte hypertrophy [45]. However, both increase AF inducibility/stability and atrial sympathetic innervation [45,46].

Ageing, arguably one of the most relevant AF risk factors, leads to increased endomysial fibrosis [47]. In a study comparing dogs at different stages of (healthy) ageing, Koura et al. demonstrated that age-related alterations in tissue structure, i.e., endomysial fibrosis and gap junction channel redistribution, were associated with increased anisotropy of conduction and likelihood for microreentry [30].

Similarly to structural remodeling, the pattern of atrial electrical remodeling can differ between patient populations and between animal models [7]. As mentioned above, AF by itself leads to a shortening of the atrial action potential duration [48]. Heart failure also leads to changes in atrial electrophysiology, both in a dog model of CHF and in HF patients [49,50]. AF-induced changes in ion channel expression differ between patient/animal models without and with (preexistent) HF showing divergences from a theoretical additive effect [49,50]. Potentially, this represents a complicating factor in the identification of atrial-selective arrhythmic drug targets (e.g., TASK-1, Kv1.5, and IKACh) in that the effects of such drugs may be affected by the presence of AF risk factors and concomitant electrophysiological changes [51].

Information about the sex of the animals included in animal studies is often not reported, creating a knowledge gap in the biological processes underlying the risk factors for atrial remodeling in males and females. Nonetheless, major risk factors for atrial cardiomyopathy differ by sex, with more hypertension, valvular heart disease, AF recurrence, and AF-related stroke in women, but more coronary artery disease and higher AF prevalence in men [52,53,54,55]. A larger number of cardiovascular risk factors in female AF patients even contributed to a worse quality of life [56]. More advanced electrical atrial remodeling or enhanced fibrotic burden have been reported in females [54,57,58,59]. Especially in the context of AF, valvular disease and aging, women have a larger quantity of atrial fibrosis [57,59].

## 4. Atrial Cardiomyopathy and Thrombogenesis

Atrial cardiomyopathy accompanying AF is associated with thromboembolic events. AF patients show increased cardiovascular mortality due to sudden death, HF and stroke. The risk of developing thromboembolic stroke increases five-fold after patients have developed AF [1,60].

The pathogenesis of thrombus formation during AF is multifactorial and results from changes in physiological processes leading to aberrant blood flow/stasis in the fibrillating atria, endothelial dysfunction/changes in endothelial structure, and hypercoagulability [61]. These mechanisms lead to the fulfillment of Virchow’s triad and predispose patients to thrombogenic events within the atria (Figure 2) [62].

### 4.1. Blood Stasis and Endothelial Dysfunction

Atrial contractile remodeling leads to reduced and/or dyssynchronous atrial contraction and wall motion disturbances. This results in blood stasis, which critically contributes to thrombogenesis.

During the first days after AF onset, loss of synchronized atrial contraction goes hand in hand with electrical remodeling processes [63]. Interestingly, although electrical remodeling is reversible upon sinus rhythm (SR) restoration, the impairment of atrial contractility partially remains after the cardioversion to SR, increasing the risk of thrombus formation and stroke [64,65,66,67].

The loss of atrial contractility contributes to thrombogenesis via multiple other mechanisms. As recently demonstrated by Spartera and colleagues, a left atrial myopathic phenotype, including reduced left atrial function, is associated with altered left atrial flow characteristics in patients at moderate-to-high risk of stroke, regardless of a history of AF [68]. In fact, altered atrial flow velocity and vorticity are expected to reduce endocardial shear stress. This phenomenon has been shown to downregulate the endothelial production of nitric oxide, which mediates vasodilation and has anti-thrombotic properties [69]. The downregulation of atrial nitric oxide would, therefore, not only stimulate the aggregation of platelets, but also increase the expression of the protein plasminogen activator inhibitor-1 (PAI-1), resulting in impaired fibrinolysis [61]. Moreover, atrial contractile dysfunction has been associated with atrial dilation, which is an independent risk factor for thrombogenesis in patients with and without AF [25,46,70,71]. In fact, atrial dilation and volume overload of the left atrial appendage are associated with increased endocardial expression of the glycoprotein von Willebrand Factor (vWF), a well-documented marker of endothelial dysfunction [72,73,74]. vWF mediates platelet adhesion to the activated endothelium, and its plasma levels are an independent predictor of poor outcome, including thromboembolic events, in patients with AF [75].

The deterioration of endothelial function in atrial cardiomyopathy can also result from inflammatory processes. As recently reviewed elsewhere, systemic and local (atrial) inflammation is a well-documented phenomenon in AF [76,77]. Within the atria, inflammation leads to areas of endothelial denudation and predisposes patients to thrombotic aggregation [78]. The exposure of tissue factor (TF)-expressing subendothelium to the bloodstream, as a consequence of endothelial denudation, may facilitate the activation of the coagulation cascade within the atrial cavity [79]. Moreover, pro-inflammatory stimuli can directly support thrombotic events by upregulating the expression of vWF and TF in endothelial cells and monocytes [80,81].

### 4.2. Pro-Thrombotic Interstitial Changes

During the complex etiology of atrial cardiomyopathy, with a variety of molecular and structural changes taking place in the atrial tissue, pro-thrombotic and pro-inflammatory changes may also be observed within the interstitial space of the atrial myocardium itself (Figure 2). For example, the accumulation of epicardial adipose tissue (EAT) may play a role in the development of AF. Several studies have reported that EAT volume may represent an independent risk factor for AF development and a predictor of AF recurrence in patients undergoing AF ablation [82,83]. The exact role of EAT in AF development still requires clarification. As reported by Antonopoulos and colleagues, EAT may play a protective role in the heart by decreasing myocardial oxidative stress via the secretion of adiponectin [84].

Nevertheless, EAT is associated with fatty infiltration from the epicardial layer, which may cause disorganized conduction within the atria [85]. Moreover, both EAT and fatty infiltration are active sources of pro-inflammatory cytokines (e.g., monocyte chemoattractant protein-1 (MCP-1), Interleukin-6 (IL-6), and tumor necrotic factor-alpha (TNF-α), which can aggravate the effect of existing pro-inflammatory processes on the endocardial endothelium and support the infiltration of immune cells within the myocardium [8].

For example, increased macrophage infiltration has been observed in the atrial myocardium of patients with atrial fibrillation [86]. Sun et al. showed that AF may further contribute to the polarization of this cell type into a pro-inflammatory phenotype, characterized by the increased expression of IL-1β [87].

Due to their proximity to cardiac cells, macrophages are also important mediators of electrical and structural remodeling processes, which may contribute to the pathogenesis of AF [88]. For example, macrophages can stimulate the activation and proliferation of cardiac fibroblasts [89]. Cardiac fibroblasts, activated during myocardial remodeling processes, can induce the accumulation of the extracellular matrix (ECM) and fibrosis, but may also support inflammation by releasing pro-inflammatory mediators [90].

Finally, the association between AF and stroke might also be due to the common causes and shared etiologies of both conditions. For example, coronary artery disease can lead to ischemia in the atrial myocardium. AF can also lead to atrial supply demand ischemia, as described in a pig model [11]. Ischemic conditions can enhance the interstitial expression and activity of coagulation factors in the myocardium. In a rabbit model of ischemia-reperfusion injury, both TF mRNA levels and pro-coagulant activity were increased in the at-risk ischemic regions of the myocardium compared to control [91]. This, potentially in combination with vascular leakage, resulting from inflammatory pathways that compromise the normal vascular function, may create favorable conditions for the activation of the extravasated coagulation factors within the myocardial interstitial space during AF [92]. 

### 4.3. Hypercoagulability

Another mechanism that contributes to thrombogenesis in AF and in other atrial cardiomyopathies consists of alterations in blood constituents which confer a hypercoagulable state [93].

Hypercoagulability in AF patients is often reflected by increased systemic platelet activation, elevated concentrations of pro-thrombotic indices (e.g., prothrombin fragments 1 + 2 and thrombin–antithrombin complex) and altered fibrinolytic activity [61].

Interestingly, the activation of the coagulation system in AF may not be homogeneous throughout the body. Some studies have shown that platelet activation and thrombin generation markers were elevated in the atria of patients within minutes after AF induction, or as a consequence of rapid atrial pacing (RAP) in animal models, compared to peripheral circulation [94,95]. These data highlight the effect of rhythm and rate on atrial pro-thrombotic mechanisms (e.g., local endocardial dysfunction/damage), which may promote a local pro-thrombotic environment.

Nevertheless, it is still not fully clarified whether “lone AF” (AF in the absence of apparent comorbidities) is sufficient to cause a pro-thrombotic state, or whether the presence of other underlying comorbidities and risk factors is required.

In a study on young very-low-risk patients with paroxysmal AF, no hypercoagulable state was observed. However, these patients did have increased levels of Factor (F) IXa–antithrombin complexes, suggesting the presence of a pre-thrombotic state [96]. Due toof this finding, the authors suggested that a second pro-coagulant hit may be needed to provoke a systemic pro-thrombotic response during AF. Clinical studies in AF patients have shown that the stroke risk often increases gradually over many years, with ageing and the development of other comorbidities, such as heart failure [97,98].

From this point of view, diabetes mellitus is a well-established risk factor for stroke in patients with AF [1]. The risk of stroke appears to increase with the increase in the duration of this metabolic disorder [99]. The impairment of vascular function (e.g., altered nitric oxide vasodilation), increased early-age arterial stiffness and systemic inflammation are well-known hallmarks of diabetes, which are also considered relevant mechanisms predisposing patients to thrombogenic events. Furthermore, as recently reviewed by Li and colleagues, diabetes is associated with a pro-thrombotic state attributable to increased platelet reactivity, quantitative alterations of coagulation factors, and hyperfibrinolysis [100].

Elderly people show a subtle increase in plasma levels of pro-coagulant factors, sometimes accompanied by a decrease in anti-coagulation and fibrinolytic factors [101].

In other words, coagulation enzyme activity seems to be age-dependent, which may explain the additional stroke risk during AF.

Unpublished data from our group indicate that advanced age and AF synergistically increase thrombin generation potential in goats with four weeks of atrial fibrillation, suggesting that advanced age may serve as a “second pro-coagulant hit” and may potentiate the effect of AF, leading to a pro-thrombotic state.

## 5. Activation of Coagulation Supports Atrial Cardiomyopathy

Few studies have investigated the possible effect of activated coagulation factors on cardiac remodeling, although this may be crucial to unravel the pathophysiology of atrial cardiomyopathies such as AF.

Activated coagulation factors, such as thrombin and FXa, can modulate physiological and pathological processes, such as inflammation and fibrosis, which may contribute to atrial cardiomyopathy (Figure 3) [102,103]. These extravascular (non-hemostatic) functions impact different cell types (e.g., endothelial cells, cardiomyocytes and cardiac fibroblasts) via the activation of protease-activated receptors (PAR) [104]. The PAR family consists of four isoforms (PAR-1 to −4). Activated coagulation proteases, such as thrombin and FXa, cleave PAR at the N-terminus and generate an exposed N-tethered ligand that self-activates the receptor [105].

Recently, hypercoagulability has been described to play a role in the progression of AF [106]. The in vivo inhibition of FXa attenuated AF-induced atrial endomysial fibrosis and reduced AF complexity in goats after four weeks of AF [106]. Several other studies have reported that the direct FXa inhibitor, rivaroxaban, attenuated cardiac fibrosis in various animal models of myocardial remodeling [34,107,108,109].

To understand the mechanisms responsible for these effects, we investigated the direct effect of activated coagulation factors, thrombin and FXa, on primary cardiac fibroblasts (CFs) [110]. In this study, thrombin and FXa lead to the increased expression of well-known pro-fibrotic genes (e.g., Alpha 2 smooth muscle actin and Transforming growth factor beta genes) in CFs. Furthermore, FXa upregulated the gene expression of two key regulators of inflammatory processes, CCL2 and IL6, in primary adult human atrial CFs. This effect was mainly caused by FXa-induced PAR-1 activation, which was the most abundant isoform in CFs. Moreover, in line with previous findings, we provided evidence for the existence of a positive feedback loop of PAR expression upon their activation by these coagulation factors [106,110].

Finally, FXa can also play a role in cardiomyocyte hypertrophy. Unpublished data from our group have shown that FXa inhibition, via rivaroxaban treatment, prevented AF-related atrial myocyte hypertrophy in a goat model of persistent AF. Myocyte hypertrophy can be induced by different stimuli during AF. One example is atrial stretch as a consequence of AF-related increased atrial pressure. However, the protective effect of FXa inhibition in our study suggests that activated coagulation factors (e.g., FXa and thrombin) may directly or indirectly contribute to myocyte hypertrophy. This observation agrees with a recent report by Guo et al., who described that exposure of rat neonatal cardiomyocytes to FXa induced hypertrophy and the increased expression of NPPA (atrial natriuretic peptide) via the activation of either PAR-1 or PAR-2 [107]. Further research is needed to elucidate the contribution of activated coagulation factors to remodeling processes such as inflammation, fibrosis and hypertrophy in atrial cardiomyopathy.

## 6. The Complex Association of AF and Thrombogenesis (Stroke)

In recent years, the interaction between AF and stroke has been shown to be far more complicated than initially believed. The traditional hypothesis was that AF causes a reduction in the blood flow velocity, activation of coagulation factors in the blood and endothelial remodeling that in combination explain the enhanced risk for stroke in patients with AF. This hypothesis explains the association between AF and stroke largely by monodirectional causation from comorbidities to AF, to the activation of coagulation factors and ultimately to stroke (Figure 4, left).

The substudy of the ASSERT trial by Bambatti et al. was the first to demonstrate that many strokes occur not only during the weeks and months after an AF episode but also before AF has occurred [111]. Additionally, genome-wide association studies have demonstrated that several common gene variants that are associated with AF are also associated with strokes, independently from the actual presence of AF. Finally, multiple pleiotropic effects of coagulation factors have been identified in various tissues over the past 10 years, including prohypertrophic, proinflammatory and profibrotic effects [111]. These findings together strongly suggest that the causal interaction between AF and stroke is multidirectional. Indeed, experimental studies suggest that the activation of coagulation factors can contribute to cardiac fibrosis and may enhance the propensity to AF [106].

In addition, AF and stroke might share common pathophysiological pathways, which may explain the association between the two entities due to the existence of a common underlying cause. In recent years, evidence has emerged that many comorbidities cause complex pathophysiological changes in the atrial wall, an atrial cardiomyopathy, that exposes the patients to an increased risk of stroke, as well as to an increased risk of AF.

For example, coronary artery disease causes ischemia in the atrial wall, which is known to increase the expression of TF on myocytes [91]. The resulting activation of interstitial coagulation factors can activate fibroblasts and result in trans differentiation into myofibroblasts and the production of collagen fibers. Fibrosis, in turn, can promote conduction disturbances and AF [9,19]. Another example is heart failure, which on the one hand can cause atrial dilatation and loss of atrial contractility, contributing to a thrombogenic environment by reducing shear stress along the atrial wall [112]. On the other hand, atrial dilatation due to heart failure causes atrial stretch, which can activate the local renin-angiotensin system that in turn can cause fibrosis through the activation of fibroblasts [41]. In both scenarios, the proarrhythmic and prothrombotic mechanisms share common pathophysiological pathways, which at some point cross the atrial endothelium, resulting in proarrhythmic mechanisms ultimately leading to AF, and at the same time, prothrombotic mechanisms promoting thrombus formation potentially causing strokes.

In summary, we believe that the association between AF and stroke is due to a multidirectional causal relation between atrial cardiomyopathy and hypercoagulability, which provoke each other but also share common underlying causes.

## Figures and Tables

**Figure 1 cells-11-02963-f001:**
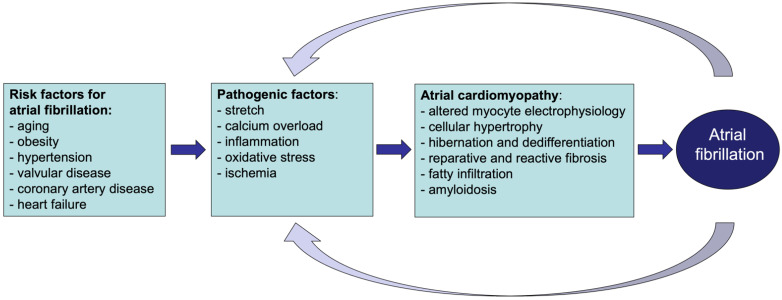
**Schematic representation of the relation between AF and atrial cardiomyopathy.** Risk factors for AF lead to pathological structural and functional changes in the atria. These result in atrial cardiomyopathy of which AF is its most relevant clinical manifestation. Once AF develops, it supports and accelerates the ongoing pathological changes in the atria.

**Figure 2 cells-11-02963-f002:**
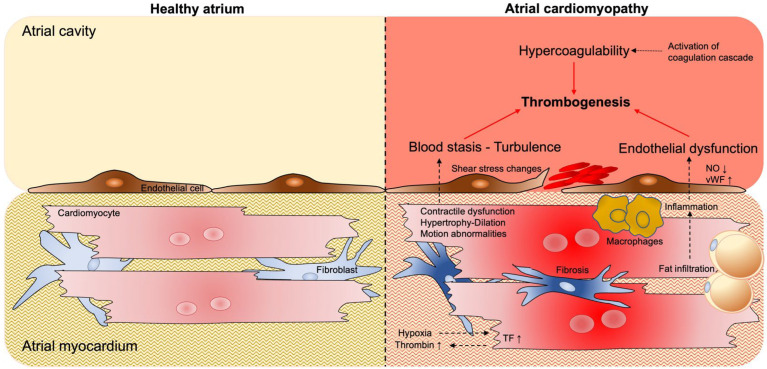
**Atrial cardiomyopathy contributes to thrombogenesis.** Unlike in the healthy atrium (**left** side), in the cardiomyopathic atrium (**right** side), pathological structural and functional changes (e.g., contractile dysfunction, atrial dilation, fibrosis, and fat infiltration) lead to aberrant blood flow and stasis in the atria cavity, endothelial dysfunction (and structural changes), and hypercoagulability, predisposing patients to thrombogenic events within the atrial cavity. Furthermore, hypoxic conditions, together with vascular leakage, may contribute to the activation of the coagulation cascade within the myocardial tissue. Abbreviations: NO = nitic oxide; vWF = von Willebrand factor; TF = tissue factor.

**Figure 3 cells-11-02963-f003:**
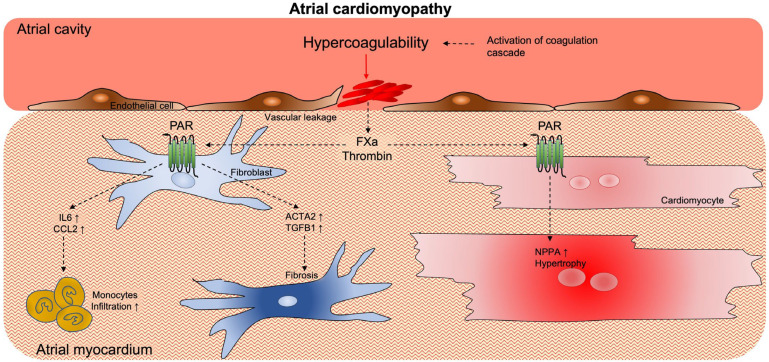
**Activation of coagulation promotes atrial cardiomyopathy.** Activated coagulation factors, such as Thrombin and FXa, modulate cellular processes via the activation of PAR expressed on cardiac cells. These processes, such as inflammation, fibrosis and cellular hypertrophy, may contribute to the worsening of atrial cardiomyopathy. Abbreviations: PAR = protease-activated receptor; FXa = Factor × activated; IL6 = Interleukin 6; CCL2 = C-C motif ligand 2; NNPA = atrial natriuretic peptide.

**Figure 4 cells-11-02963-f004:**
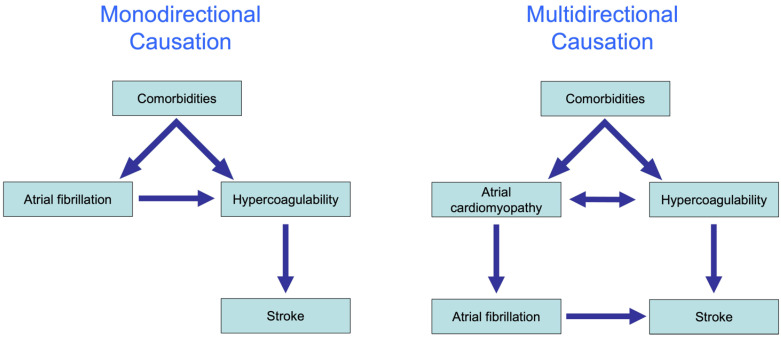
**The complex association between AF and stroke.** Monodirectional causation (**left**): various comorbidities lead to the onset of AF, followed by the activation of the coagulation system, and ultimately stroke. Multidirectional causation (**right**): atrial cardiomyopathy and hypercoagulability cause each other and share common pathophysiological pathways. These pathways, which may occur within and/or outside the atrial endothelium, can contribute to both proarrhythmic and prothrombotic mechanisms, resulting in the concomitant increased risk of AF and stroke.

**Table 1 cells-11-02963-t001:** Animal studies on disease entities associated with an increased risk of AF in patients.

Disease in Humans	Intervention in Animal Model	Species	Max. Duration	Main Feature of Structural Remodeling	Reference
**Lone AF’**	Rapid atrial pacing	Goat/dog/sheep	6 months	Myocyte hypertrophy and endomysial fibrosis	Verheule, Circ AE, 2013 [20]
**Vagal AF**	Acetylcholine administration	Dog/sheep perfused atria	Seconds	None; acute model	Schuessler, Circ Res 1992 [21]
**Post-operative AF**	Sterile pericarditis	Dog	3–4 days	Gap junction redistribution	Ryu, Am J Physiol Heart Circ Physiol, 2007 [22]
**Congestive heart failure**	Rapid ventricular pacing	Dog	5 weeks	(Replacement) fibrosis	Li, Circ, 1999 [23]
**Valvular insufficiency**	Mitral valve avulsion	Dog	4 weeks	Heterogeneous fibrosis	Verheule, Circ, 2003 [24]
**Bi-atrial dilation**	AV node ablation	Goat	4 weeks	Myocyte hypertrophy	Neuberger, Circ, 2005 [25]
**Coronary artery disease**	RA artery ligation	Dog	8 days	Granulation tissue, replacement fibrosis	Nishida, Circ, 2011 [26]
**Obesity**	High-fat diet	Sheep	8 months	Fibrosis, adipocyte infiltration	Abed, Heart Rhythm, 2013 [27]
**Sleep apnea**	Tracheal occlusion	Pig	2 min	None; acute model	Linz, Heart Rhythm, 2011 [28]
**Hypertension**	Prenatal corticosteroid exposure	Sheep	4 years	Myocyte hypertrophy, myolysis, heterogenous fibrosis	Kistler, Eur Heart J, 2006 [29]
**Ageing**	Wait	Dog	8 years	Endomysial fibrosis and gap junction redistribution	Koura, Circ, 2002 [30]

Note: This table summarizes the most relevant studies in which disease entities, which are associated with an increased risk of AF in patients, have been investigated in animal models. Abbreviations: AV = atrioventricular; RA = right atrium.

## Data Availability

Not applicable.

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
