# Peer review of "The Complex Relation between Atrial Cardiomyopathy and Thrombogenesis"

_cells, 2022, doi:10.3390/cells11192963_

Round 1

Reviewer 1 Report

Manuscript ID: cells-1859757

Type of manuscript: Review

Title: The complex relation between atrial cardiomyopathy and thrombogenesis

Authors: Elisa D'Alessandro, Joris Winters, Frans A Van Nieuwenhoven, Ulrich Schotten, Sander Verheule * Submitted to section: Cells of the Cardiovascular System

In the review article cells-1859757 entitled “The complex relation between atrial cardiomyopathy and thrombogenesis” Elisa D'Alessandro et al. target a very interesting topic  - the atrial cardiomyopathy - which is currently less understood and need to be elucidated.

The authors summarize here some important mechanisms of atrial cardiomyopathy and address the complex association between AF and stroke.

The context of the pathophysiology of atrial cardiomyopathy is not yet understood. Based on this a review article in this field have also to be underly al the gaps of knowledge in the field and can only summarize the existing knowledge. According to this, the article is really short and showed be extended for more important aspects.

However, there are some important aspects missing:

ð  One major issue: The authors hypothesize the mechanism of: AF induce atrial cardiomyopathy – why? Please discuss the way atrial cardiomyopathy induce AF! In a lot of patients (e.g. DCM, HCM etc.) the atrial alterations are first and induced by heterocellularity subsequently the atrial arrhythmia – and is than in the atrial arrhythmopathy status - with the clinical phenotype of atrial fibrillation. Please modify for this absolutely important aspect your Figure 1 and include this in your review discussion.

Main comments:

-          Fig. 1 shows “Schematic representation of the relation between AF and atrial cardiomyopathy.”

ð  Why think the authors that AF is first and induced atrial cardiomyopathy? In a lot of patients, the atrial cardiomyopathy is first and pathophysiological inductor of atrial arrhythmopathy with the phenotype of AF! Please include this aspect in your article and discuss the pathophysiological context.

ð  In general Fig. 1 is to simplified for the complex context and showed include more important aspects

-          Please include and discuss (in the tbl. 1 and text) some more cellular electrophysiological alterations in important translational large animal AF model – e.g. publications by Wiedmann F/Schmidt C et al. (Heidelberg) – Biofeedback induced AF with preference of endogenous AF with ion channel alterations e.g. atrial selective ion channels e.g. TASK-1 (KCNK3) etc.

-          The authors should also discuss mechanical aspects of the atria as inductor of thrombogenesis

-          In general, the authors should include more references of publications in this field

-          Discuss the role of macrophages - and include this cell type in your Fig. 2

Reviewer 2 Report

D'Alessandro et al. summarize the complex relationship between atrial fibrillation, thrombogenesis, and stroke in their review article. The topic is interesting and novel, the MS is easy to follow and the figures are of good quality and support the text. I have only a few comments and suggestions for the authors.

Major comments:

1. The authors mention systemic metabolic alterations in the Abstract and Introduction parts. What is the relationship between type 2 diabetes mellitus and AF? The molecular mechanisms of how obesity (or DM) could influence thrombogenesis and its relationship with AF are really short in the MS. Please discuss it in more detail if it is possible.

2. It would be interesting to summarize the sex-based differences in the development and molecular mechanisms of AF and thrombogenesis. Which sex of animals were used in the studies listed in Table 1? Please add this information to Table 1 in a separate column.

Minor comment:

Full names of the abbreviations are missing in the Figure legends. 

Round 2

Reviewer 1 Report

The authors have taken into account the noted aspects of the pathogenesis of atrial cardiomyophythia and integrated them adequately into their article.

Minor:

A few words on atrial selective ion channels like TASK-1 or Kv1.5 would be good.
